# Neural Network Adaptive Quantization based on Bayesian Deep Learning

## Abstract

We propose a novel approach to solve the adaptive quantization problem in neural networks based on epistemic uncertainty analysis. The quantized model is treated as a Bayesian neural network with stochastic weights, where the mean values are employed to estimate the corresponding weights. Standard deviations serve as an indicator of uncertainty and the number of corresponding bits — i.e., a larger number of bits indicate lower uncertainty, and vice versa. We perform an extensive analysis of several algorithms within a novel framework for different convolutional and fully connected neural networks based on open datasets demonstrating the main advantages of the proposed approach. In particular, we introduce two novel algorithms for mixed-precision quantization. Quantile Inform utilizes uncertainty to allocate bit-width across layers, while RandomBits employs stochastic gradient-based optimization techniques to maximize the full likelihood of quantization. Using our approach, we reduce the average bit-width of the VGG-16 model to 3.05 with the 90.5% accuracy on the CIFAR-10 dataset compared to 91.9% for the non-quantized model. For the LeNet model trained on the MNIST dataset, we reduce the average bit-width to 3.16 and achieve 99.0% accuracy, almost equal to 99.2% for the non-quantized model.

## 1 Introduction

Compression of neural network models are typically addressed using pruning (Blalock et al., 2020), quantization (Gholami et al., 2021), decomposition of weight matrices into factors (Sainath et al., 2013) and distillation (Hinton et al., 2015) technique. If the architecture of a model is fixed, distillation cannot be applied, while pruning and decomposition of weight matrices can lead to certain complications. The quantization approach may be the only promising way to compress a model.

Quantization can be applied during (Courbariaux et al., 2015) and after model training (Fang et al., 2020), as well as during retraining (Nagel et al., 2021). According to the weights manipulation approach, these methods can be divided into several classes. One can either manipulate weight values or change the number of bits of a particular weight. In our work we focus on the latter class, where different granularity levels can be implemented. One can quantize the entire model (Yang et al., 2019), each layer (Wang et al., 2020; Xiao et al., 2022), or each channel (Chmiel et al., 2020; Zhong et al., 2020). The smaller the granularity, the better the quantization. Layered and channeled granularity levels require adaptive quantization methods (Langroudi et al., 2021) that are able to determine which weights need more bits and which ones need fewer.

Most existing methods of quantization are based on heuristic criteria or deterministic optimization, which may not capture the uncertainty and variability of network parameters (Zhou et al., 2018; Guo, 2018). Moreover, most methods such as TWN (Li et al., 2016), LR-Net (Shayer et al., 2017), RQ (Louizos et al., 2018), PACT (Choi et al., 2018) and Dorefa (Zhou et al., 2016) use a fixed bit-width for all network layers, ignoring the potential benefits of applying different bit-widths to different layers (Liang et al., 2021). A flexible framework for quantization that could account for the probabilistic nature of a network parameters and adapt the bit-width to a layer characteristics is desirable (Cheng et al., 2017; Garg et al., 2021).

Among the approaches, a group of reinforcement learning-based algorithms, such as (He et al., 2018), have been proposed for adaptive quantization. However, these methods are computationally

expensive, making them less suitable for practical applications where efficiency is crucial. Therefore, they have been excluded from consideration.

To address the challenge of the adaptive quantization problem, we propose two novel algorithms based on Bayesian neural networks (BNNs), which can capture the uncertainty of the network parameters and outputs, referred further as Quantile Inform and Random Bits. Quantile Inform is an informativeness-based algorithm that defines bit-widths of each layer based on informativeness calculated using mean values and standard deviations of Bayesian neural network parameters. Random Bits is based on BNN post-training in terms of maximization of the likelihood of optimal quantization with a limit on the average bit-width of a quantized model.

We evaluate proposed algorithms on several benchmark datasets and show that they achieve comparable or better accuracy than the existing quantization methods while using fewer bits. We show that informativeness based on weights of a trained BNN strongly correlates with the optimal bit-width for layers. Different methods of aggregation and smoothing of informativeness are evaluated in a few benchmarks and the best one is used in Quantile Inform to calculate bit-widths. Alternative informativeness values based on variances are tested as well. We also demonstrate the advantages of BNN for quantization, such as robustness to noise and calibration of uncertainty.

## 2 RELATED WORK

BNNs are able to explicitly model uncertainty, thus may be also suitable for adaptive quantization. There is a number of papers where quantization relies either on the measure of uncertainty obtained as a result of model training or on the application of uncertainty estimation to the weights of a trained model. We employ some of these methods as described below.

Blundell et al. (2015) experimentally confirmed the possibility of using averages and variances obtained by BNN training to assess the informativeness of network weights and the effectiveness of pruning (zeroing) the least informative ones. It is applied to research pruning of a trained network based on the magnitude of the signals. The authors defined the signal as the ratio of the average to the variance. It is argued that the smaller the ratio, the less meaningful the weight. Accordingly, after calculating the signals for all weights, the authors zeroed out the weights with the smallest signals. Such an experiment on the MNIST classification problem using an architecture of two fully connected layers confirmed that it is possible to zero 75% of the weights without degradation of the error value on the test subset. At zeroing 95%, there is a degradation from 1.24% to 1.29%, and at 98%, the error reaches 1.39%. Consequently, 48,000 of the original 2.467 million weights remain non-zero.

A few researchers propose a method for neural network compression based on quantization of weights using a mixture of Gaussian distributions. This approach allows the model to be trained in such a way that the weights tend to move towards one of the $K$ centroids, forming $K$ clusters which is vector quantization (Gong et al., 2014; Park et al., 2017; Stock et al., 2019; Razani et al., 2021).

After the weights of the model form sufficiently distinct clusters, they can be replaced by the corresponding centroids of these clusters, and the centroids themselves can be encoded with $\log_2 K$ bits. Moreover, if one of the centroids is zero, corresponding weights do not have to be stored or considered in calculations. This method agrees with the principle of minimum description length (MDL) proposed by Grünwald (2007), which states that the best model for data is the one that requires the smallest number of bits to describe it. According to Louizos et al. (2017), the most effective way to solve the problem of compression and computational efficiency in deep learning is the Bayesian approach. The latter states that it is possible to remove a large number of weights or neurons in a neural network using a prior distribution that encourages sparsity.

There is an innovative approach (Van Baalen et al., 2020) to decompose a particular weight into terms that determine the final bit-width of the quantized weight. This decomposition is individual for each weight and is trained using the variational Bayesian inference (Chappell et al., 2008). Thus, using Bayesian methods, the most probable number of bits for each weight is determined. Each of these terms is a learnable parameter and reflects the confidence of the algorithm that the corresponding weight should have at least the bit-width with the associated matching parameter. The above

approach ensures a higher accuracy with a better quantization of weights compared to other models. However, the limitation is that the bit-width has to be the powers of 2 (i.e. 2, 4, 8, 16, or 32 bits).

These results suggest that BNNs and approaches to their training open up new opportunities for adaptive quantization based on the measure of uncertainty, a side-product of Bayesian network training, which is leveraged by all our algorithms to measure uncertainty.

## 3 QUANTIZATION DETAILS

### 3.1 QUANTIZATION OF BNNS

A common method for training BNNs is Variational Inference (Kingma et al., 2015). Suppose we train a BNN model $\mathcal{M}$ on a dataset $\mathcal{D}$ via Variational Inference. The model $\mathcal{M}$ has $N$ layers, i.e. $\mathbf{w} = l_1 \cup l_2 \cup ... \cup l_N$, where each layer $l_i$ is treated as a set of its weights, with $l_i \cap l_j = \emptyset$ at $i \neq j$. Each weight $w \in \mathbf{w}$ of the model $\mathcal{M}$ is a Gaussian random variable $w \sim \mathcal{N}(\mu_w, \sigma_w)$. We can utilize it to construct a conventional neural network by replacing each stochastic weight $w$ with a certain real value. In this transformation, an attractive choice for each weight $w$ is its mean value $\mu_w$. Therefore, further layer quantization of the trained $\mathcal{M}$ into specified bit-widths $b_1, ..., b_N$ is performed as follows: each stochastic weight $w$ is replaced by its average value $\mu_w$, resulting in $\mathcal{M}$det, which is a deterministic version of $\mathcal{M}$. Then, every $j$-th layer of $\mathcal{M}$det is quantized using $b_j$ bits.

### 3.2 UNIFORM QUANTIZATION

We perform layer by layer quantization of $\mathcal{M}_{\text{det}}$. To do this, we select a layer of this model and call it $l_j$. Notably, for the BNN model $\mathcal{M}$, the layer $l_j$ is a set of stochastic weights $\{w\}$, while for the deterministic model $\mathcal{M}_{\text{det}}$, the layer $l_j$ is a set of average weight values $\{\mu_w\}$. Let us set the quantization interval $[\alpha_j, \beta_j]$ for the layer $l_j$. Then, we apply the quantization layer $l_j$ into $b_j$ bit as follows:

$$Q(\mu_w, b_j) = \alpha_j + s(b_j)\text{clip}\left(\frac{\mu_w - \alpha_j}{s(b_j)}\right), \quad s(b_j) = \frac{\beta_j - \alpha_j}{2^{b_j} - 1},$$

where the function $\text{clip}(x) = \text{clamp}(\text{round}(x), 0, 2^{b_j} - 1)$. The operation clamp restricts a value to lie within the range from 0 to $2^{b_j} - 1$. When using such quantization on the interval $[\alpha_j, \beta_j]$, a uniform grid of $2^{b_j}$ nodes is specified and each $\mu_w \in l_j$ is rounded to the nearest grid point. For each layer $l_j$ we consider the boundaries of the quantization interval $\alpha_j$ and $\beta_j$ as the 5th percentile and the 95th percentile of the set $\{\mu_w | w \in l_j\}$, respectively.

### 3.3 NON-UNIFORM QUANTIZATION: QUANTILIZATION

Quantilization offers an alternative to uniform interval quantization by leveraging quantiles. It is supposed to address a possible drawback in uniform quantization grids by ensuring an even distribution of $\{\mu_w\}$ among quantized values through the use of quantiles.

Let us consider the scenario where we aim to quantize a set of average weight values $\{\mu_w\}$ from layer $l_j$ of $\mathcal{M}_{\text{det}}$ into $m_j := 2^{b_j}$ values, i.e. into $b_j$ bit. Calculated directly from data, empirical quantiles, provide the best possible partition of $\{\mu_w\}$ into even groups. Unfortunately, they can be impractical to compute and store for large datasets. Therefore, we opt for a more efficient approach by relying on computing quantiles derived from the theoretical distribution $F_j$ that can be defined by only its type and parameters. For the sake of definiteness, we shall consider only Gaussian distributions.

Assuming the distribution represented by $F_j$ spans an entire real number line, we can establish a following relationship by defining $a_k := F_j^{-1}(k/m_j)$ for $0 < k < m_j$:

$$\mathbb{P}(W_1 \in (-\infty, a_1)) = \mathbb{P}(W_1 \in (a_1, a_2)) = \ldots = \mathbb{P}(W_1 \in (a_{m_j-1}, +\infty)).$$

This indicates that all quantization intervals will, on average, contain equal number of elements.

The next step is to decide how to quantize the $\{\mu_w\}$ from each interval $(a_{i-1}, a_i)$ into a single value $\bar{a}_i$. For finite intervals $(a_{i-1}, a_i)$, we can minimize the absolute quantization error by selecting the midpoint as $\bar{a}_i = (a_{i-1} + a_i)/2$. In case of infinite intervals $(-\infty, a_1)$ and $(a_{m_j-1}, +\infty)$, additional considerations may be necessary, particularly when dealing with outlier values. Otherwise, we can use medians $\bar{a}_1 = F_j^{-1}(1/2m_j)$ and $\bar{a}_{m_j} = F_j^{-1}((2m_j - 1)/2m_j)$. This ensures that approximately the same number of values are quantized upwards and downwards. Additional information is presented in Appendix B.

## 4 QUANTIZATION ALGORITHMS

### 4.1 WEIGHT INFORMATIVENESS

Consider a trained BNN $\mathcal{M}$, and define weight informativeness as the ratio of the absolute mean value to the corresponding variance:

$$\text{info}(w) = \frac{|\mu_w|}{\sigma_w}. \tag{1}$$

Blundell et al. (2015) showed that informativeness in pruning can preserve the quality of predictions with a high percentage of pruning. For a quantization algorithm, as in the case of the pruning criterion, one can use informativeness weights. If we employ a computationally expensive method of finding optimal bit-widths for each layer of the neural network, such as Bayesian optimization described by Snoek et al. (2012), bit-widths correlate with the informativeness of each layer. The following experiment was carried out on the CIFAR-10 dataset with VGG-7 (with an additional fully connected layer containing 10 neurons) and VGG-16 models and the MNIST dataset with LeNET model to confirm this hypothesis.

We extended the use of informativeness to pruning and developed a quantization method that assigns an optimal bit-width to each layer, proportional to its informativeness. Our analysis revealed a strong positive correlation between the optimal bit-width produced by the complete brute-force of all possible bit-widths for each layer and informativeness produced by a fitted Bayesian version of the network. All experiments proved this, see Figure 1, as measured by the coefficient of determination R2. The calculated R-measures of correlation between layer informativeness and bit-width obtained through Bayesian optimization are 0.823 for the VGG-7 model, 0.867 for VGG-16, and 0.854 for LeNet. These findings suggest that informativeness can be used as a reliable predictor of bit-width in adaptive quantization.

Analysis of experimental data revealed that the initial and final layers of the neural network play a more critical role, which is expressed in the need for higher bit resolution. It confirms the well-known heuristics to avoid the quantization of the first and last layers discussed in Louizos et al. (2017) and Gholami et al. (2021). We can try to explain this phenomenon as follows. The last layer can be regarded as a classifier based on features extracted with the first and intermediate layers of a neural network. Thus, the last layers should be sensitive, which may also be related to their capacity. Typically, the last layers are not as large as the layers in the middle of the model. It is not unlikely that only a small part of these weights is useful for solving the task. In this case, the mean bit-width of the intermediate layer should be small enough. The same reasoning applies to the first layers which usually have a significantly fewer weights than the intermediate ones. Additionally, the first layer requires a larger bit-width for quantization in order to correctly extract low-level features, which are important for constructing more complex features and training the network.

### 4.2 INFORMATION-BASED QUANTIZATION: QUANTILE INFORM

Let us denote the quantized version of the model $\mathcal{M}_{\text{det}}$ by $\mathcal{M}_{\text{quant}}$. We assume that the average bit-width of the quantized model $\mathcal{M}_{\text{quant}}$ can be no more than the desired $\gamma$ bit. At the same time, we aim to perform quantization while preserving the model's performance as much as possible.

Our algorithm based on informativeness is called Quantile Inform. Given a desired average bit-width for the network, we solve an optimization problem to find the informativeness thresholds that yield the target bit-width. The method includes the following steps, see Figure 2:

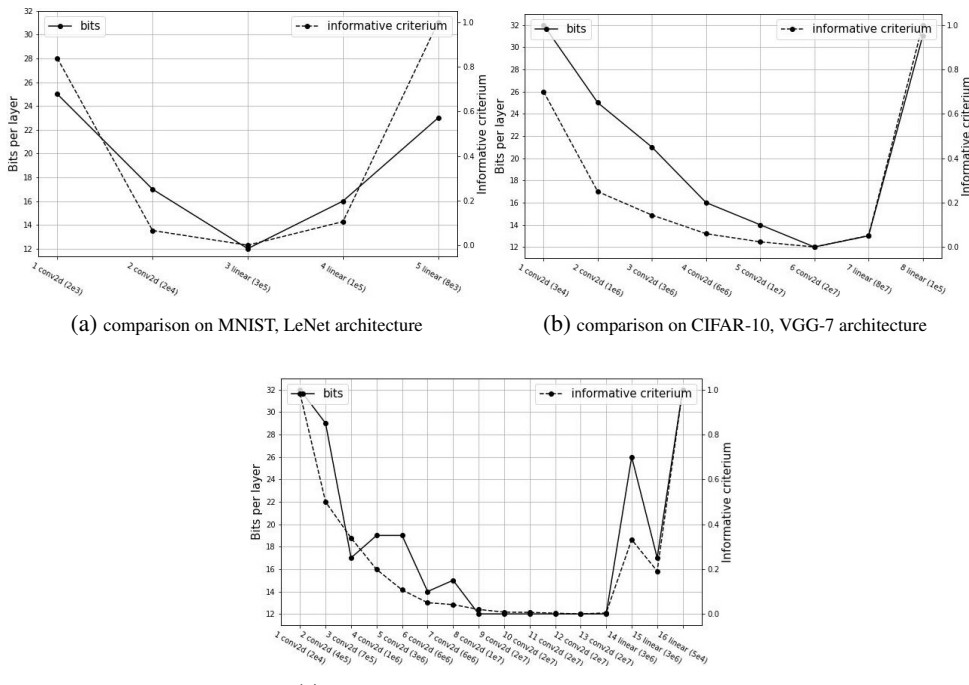

(a) comparison on MNIST, LeNet architecture      (b) comparison on CIFAR-10, VGG-7 architecture

(c) comparison on CIFAR-10, VGG-16 architecture

Figure 1: Correlation between informativeness of layers and bit-width obtained through Bayesian optimization

1) Training BNN. First, the BNN must be trained to estimate the uncertainty of predictions. This is achieved by applying Bayesian learning methods, such as Variational Inference or Monte Carlo dropout in Gal & Ghahramani (2016).

2) Informativeness calculation. After training the BNN, the informativeness of each neuron can be calculated.

3) Quantization. More important neurons have higher informativeness, which means that they contain more significant and stable activations. These neurons can be selected for preservation with full precision, whereas less informative ones can be quantized with a loss of precision. Since all weights of layer $l_j$ quantize into the same number of bits, we will aggregate the informativeness of its weights for any layer $l_j$ as follows:

$$I_j = \frac{1}{|l_j|} \sum_{w \in l_j} \text{info}(w), \tag{2}$$

where $|l_j|$ is the number of weights in the layer $l_j$. To determine the set of bits, the informativeness is normalized to $[0, 1]$: $\widetilde{I}_j \leftarrow (I_j - \min_k I_k)/(\max_k I_k - \min_k I_k)$. Next, we extract a root from normalized informativeness as $\widehat{\widetilde{I}}_j \leftarrow (\widetilde{I}_j)^{1/2}$. Consequently, $k$ thresholds are selected within the interval $[0, 1]$, where $k$ is the number of bit-widths available for the quantization process. The thresholds are evenly spaced on the range and then scaled. This allows one to obtain and test different sets of bits by changing only one scaling factor.

4) Repeated iterations. Steps 2 and 3 can be repeated several times with different thresholds to determine the bits based on informativeness, thereby achieving the desired level of quantization and accuracy.

### 4.3 RANDOM BITS

Let us consider the quantization problem in a probabilistic formulation. Suppose we have a trained BNN model $\mathcal{M}$, which consists of $N$ layers: $\mathbf{w} = l_1 \cup l_2 \cup ... \cup l_N$. Consider a certain layer $l_j$ of an $\mathcal{M}$. Each weight $w$ of this layer has a normal distribution with trained parameters $q(w) = \mathcal{N}(w|\mu_w, \sigma_w)$. Our aim is to quantize the mathematical expectations $\{\mu_w\}$ of this layer into $b_j$ bit.

Figure 2: Quantile Inform assigns bit-widths proportional to the average per-layer informativeness based on uncertainties of a fitted Bayesian neural network.

We also need to choose a quantization interval $[\alpha_j, \beta_j]$, where the quantized values are located. After quantization, we obtain the deterministic layer version $l_j$ (i.e. $w \to \mu_w$). When its weights are independent, the likelihood function will be equal to

$$p(l_j|b_j) = \prod_{w \in l_j} q(Q(\mu_w, b_j)).$$

However, all the weights in the layer are quantized into the same bit-width. For this reason, we have to reject the assumption of independence of its weights. Thus, when layer weights $l_j$ are independent, we can approximate their joint density by the linear combination:

$$p(l_j|b_j) = \sum_{w \in l_j} c_j(w)q(Q(\mu_w, b_j)), \text{ where } \sum_{w \in l_j} c_j(w) = 1, c_j(w) \geq 0.$$

As coefficients $\{c_j(w)\}$ we can take the normalized informativeness of the weights:

$$c_j(w) = \frac{\text{info}(w)}{\sum_{w_k \in l_j} \text{info}(w_k)}.$$

The likelihood of model quantization with weights when quantized in bits $\mathbf{b} = (b_1, ..., b_N)$ is expressed by the formula

$$p(\mathcal{M}_{\text{quant}}|\mathbf{b}) = \prod_{j=1}^{N} p(l_j|b_j) = \prod_{j=1}^{N} \sum_{w \in l_j} c_j(w)q(Q(\mu_w, b_j)). \tag{3}$$

Note that maximizing the expression (3) means that the quantized averages $Q(\mu_w, b_j)$ will be close to $\mu_w$. In this case, the model will maintain its accuracy. Therefore, we would like to maximize (3). One should bear in mind that bit-widths are discrete. Consequently, gradient optimization methods are not applicable in this case. To solve this issue, we propose to consider bits as random discrete values. This allows us to use gradient optimization techniques.

We will assume that the bit-width $b_j$ of each layer $l_j$ is a discrete random variable (for simplicity, we say that the maximum $b_{\text{min}}$ and the minimum $b_{\text{max}}$ quantization bits are 1 and 32, respectively):

$$b_j = \sum_{k=1}^{32} k \cdot \mathbb{I}[b_j = k], \qquad \mathbb{E}[b_j] = \sum_{k=1}^{32} k \cdot \mathbb{P}[b_j = k].$$

Let us introduce for each $b_j$ a set of trainable parameters $(\phi_{j,1}, \phi_{j,2}, ..., \phi_{j,32}) = \boldsymbol{\phi}_j$. Probability events $\{b_j = k\}$ are modeled using the Softmax function:

$$\mathbb{P}[b_j = k] = \text{Softmax}(\boldsymbol{\phi}_j)_k = \frac{e^{\phi_{j,k}}}{\sum_{i=1}^{32} e^{\phi_{j,i}}}. \tag{4}$$

Let $\mathcal{A} = \{1, 2, ..., 32\}$. Then, the distribution $p(\mathbf{b})$ of the random vector $\mathbf{b} = (b_1, ..., b_N)$ has the form

$$p(\mathbf{a}) = \mathbb{P}[\mathbf{b} = \mathbf{a}] = \mathbb{P}[b_1 = a_1, ..., b_N = a_N] = \prod_{j=1}^{N} \mathbb{P}[b_j = a_j], \quad \mathbf{a} \in \mathcal{A}^N.$$

Interpreting bits as random variables allows us to maximize the full likelihood of quantization:

$$p(\mathcal{M}_{\text{quant}}) = \sum_{\mathbf{b} \in \mathcal{A}^N} p(\mathcal{M}_{\text{quant}}|\mathbf{b})p(\mathbf{b}) = \mathbb{E}_{p(\mathbf{b})}[p(\mathcal{M}_{\text{quant}}|\mathbf{b})] \to \max_{\boldsymbol{\Phi}}, \tag{5}$$

where $\boldsymbol{\Phi} = (\phi_1, ..., \phi_N)$. Now that we let the quantization intervals $[\alpha_j, \beta_j]$ change, the function $Q$ also changes, so maximization of (5) will also include $\boldsymbol{\alpha} = (\alpha_1, ..., \alpha_N), \boldsymbol{\beta} = (\beta_1, ..., \beta_N)$.

Instead of maximizing (5), the following minimization problem can be solved (see Appendix A.1)

$$\sum_{j=1}^{N} \sum_{i=1}^{32} \tau_{j,i} \mathbb{P}[b_j = i] \to \min_{\boldsymbol{\Phi}, \boldsymbol{\alpha}, \boldsymbol{\beta}}, \tag{6}$$

where coefficients $\{\tau_{j,i}\}$ are calculated as follows:

$$\tau_{j,i} = \sum_{w \in l_j} c_j(w) \frac{(Q(\mu_w, b_j = i) - \mu_w)^2}{\sigma_w^2}. \tag{7}$$

Note that this problem has a solution when for each $j$ probability $\mathbb{P}[b_j = 32] \to 1$. Let us define the mean bit-width of the quantized model as

$$b_{\text{mean}} = \frac{1}{|W|} \sum_{j=1}^{N} b_j |l_j|, \tag{8}$$

where $|l_j|$ is the number of weights in the j-th layer. If we have the desired average bit-width $\gamma$ of the quantized model, we can add L2 penalty between $b_{\text{mean}}$ and $\gamma$. Then, we get the following loss function

$$\mathcal{L}(\boldsymbol{\Phi}) = \overbrace{\sum_{j=1}^{N} \sum_{i=1}^{32} \tau_{j,i} \mathbb{P}[b_j = i]}^{\text{quantization error}} + \underbrace{\eta (\mathbb{E}_{p(\mathbf{b})}[b_{\text{mean}}] - \gamma)^2}_{\text{regularizer}} \to \min_{\boldsymbol{\Phi}, \boldsymbol{\alpha}, \boldsymbol{\beta}}, \tag{9}$$

where the first term is responsible for the quality of quantization, and the second one is for the proximity between $b_{\text{mean}}$ and $\gamma$.

The loss function (9) has computable derivatives with respect to the parameters $\boldsymbol{\Phi}$, but the derivatives with respect to the boundaries $\boldsymbol{\alpha} = (\alpha_1, ..., \alpha_N)$ and $\boldsymbol{\beta} = (\beta_1, ..., \beta_N)$ of the quantization intervals are extremely complex. Therefore, we introduce a second loss function to calculate these derivatives.

Let us assume that we know the specific parameters $\boldsymbol{\Phi}$. Then we can express quantization bits as rounded mathematical expectations of random bits:

$$\widehat{b}_j = \text{round}(\mathbb{E}_{p(b_j)}[b_j]) = \text{round}\left( \sum_{k=1}^{32} k \cdot \mathbb{P}[b_j = k] \right), \tag{10}$$

where the probabilities $\mathbb{P}[b_j = k]$ are calculated via (4). In this case, we can rewrite the conditional likelihood (3) expressed in terms of quantization bits $\widehat{\mathbf{b}} = (\widehat{b}_1, ..., \widehat{b}_N)$:

$$p(\mathcal{M}_{\text{quant}}|\mathbf{b} = \widehat{\mathbf{b}}) = \prod_{j=1}^{N} p(l_j|\widehat{b}_j) = \prod_{j=1}^{N} \sum_{w \in l_j} c_j(w) q\left( Q(\mu_w, \widehat{b}_j) \right) \to \max_{\boldsymbol{\alpha}, \boldsymbol{\beta}}. \tag{11}$$

The solution of (11) leads to the adaptation of the quantization intervals boundaries $\boldsymbol{\alpha}$ and $\boldsymbol{\beta}$ to the found quantization bits $\widehat{b}_1, ..., \widehat{b}_N$. Instead of maximizing the expression (11), we can minimize the following loss function (see Appendix A.2)

$$\mathcal{T}(\boldsymbol{\alpha}, \boldsymbol{\beta}) = \sum_{j=1}^{N} \sum_{w \in l_j} c_j(w) \frac{(Q(\mu_w, \widehat{b}_j, \alpha_j, \beta_j) - \mu_w)^2}{\sigma_w^2}. \tag{12}$$

The derivatives of $\mathcal{T}(\boldsymbol{\alpha}, \boldsymbol{\beta})$ with respect to boundaries $\boldsymbol{\alpha}, \boldsymbol{\beta}$ can be approximated as follows (see Appendix A.3):

$$\frac{\partial \mathcal{T}}{\partial \beta_j} \approx \frac{2}{\beta_j - \alpha_j} \sum_{w \in l_j} c_j(w) \frac{(Q(\mu_w, \widehat{b}_j) - \mu_w)^2}{\sigma_w^2}, \quad \frac{\partial \mathcal{T}}{\partial \alpha_j} \approx -\frac{\partial \mathcal{T}}{\partial \beta_j}. \tag{13}$$

The algorithm works in the following manner. At each iteration of the algorithm, the first loss function is calculated using (9). The quantization bits $\widehat{\boldsymbol{b}} = (\widehat{b}_1, ..., \widehat{b}_N)$ and mean model bit $b_{\text{mean}}$ are then calculated via (10) and (8). Next, if $b_{\text{mean}}$ is less than the desired model mean bit $\gamma$, the quantization interval boundaries are updated using gradient minimization of the second loss function (12). Let us denote the number of boundary updates as $\mathcal{K}$.

---

**Algorithm 1:** Random Bits

  **Input:**
    trained BNN model $\mathcal{M}$
    desired quantization mean bit $\gamma$
    number of boundary update iterations $\mathcal{K}$
  **initialize:**
    $\boldsymbol{\Phi} \leftarrow (\phi_1, ..., \phi_N)$
    $\boldsymbol{\alpha} \leftarrow (\alpha_1, ..., \alpha_N)$
    $\boldsymbol{\beta} \leftarrow (\beta_1, ..., \beta_N)$
  **repeat**
    recalculate $\{\tau_{j,i}\}$ coefficients using (7)
    compute first loss $\mathcal{L}(\boldsymbol{\Phi})$ via (9)
    calculate bits $\widehat{\boldsymbol{b}} = (\widehat{b}_1, ..., \widehat{b}_N)$ using (10)
    get mean quantization bit $b_{\text{mean}} \leftarrow \frac{1}{|W|} \sum_{j=1}^{N} |l_j| \widehat{b}_j$
    **if** $b_{\text{mean}} < \gamma$ **then**
      set $\mathcal{T}_{best}$ equal to $+\infty$
      **for** iteration **in range**($\mathcal{K}$) **do**
        calculate second loss $\mathcal{T}_{curr} \leftarrow \mathcal{T}(\boldsymbol{\alpha}, \boldsymbol{\beta}, \widehat{\boldsymbol{b}})$ via (12)
        **if** $\mathcal{T}_{curr} < \mathcal{T}_{best}$ **then**
          save best result $\boldsymbol{b}_{best}, \boldsymbol{\alpha}_{best}, \boldsymbol{\beta}_{best} \leftarrow \widehat{\boldsymbol{b}}, \boldsymbol{\alpha}, \boldsymbol{\beta}$
        **end if**
        calculate gradients $\nabla_{\boldsymbol{\alpha}, \boldsymbol{\beta}} \mathcal{T}(\boldsymbol{\alpha}, \boldsymbol{\beta}, \widehat{\boldsymbol{b}})$ via (13)
        update
          $\boldsymbol{\alpha} \leftarrow \boldsymbol{\alpha} - \theta \nabla_{\boldsymbol{\alpha}} \mathcal{T}$
          $\boldsymbol{\beta} \leftarrow \boldsymbol{\beta} - \theta \nabla_{\boldsymbol{\beta}} \mathcal{T}$
      **end for**
    **end if**
    update parameters $\phi \leftarrow \phi - \tau \nabla_{\boldsymbol{\Phi}} \mathcal{L}(\boldsymbol{\Phi})$
  **until** loss $\mathcal{L}(\boldsymbol{\Phi})$ converges

---

# 5 EXPERIMENTS

In this section, we present the results of our experiments for the LeNet, VGG-7, VGG-16 and ResNet-20 models in the classification problems on the CIFAR-10 and MNIST datasets, and for multilayer perceptron for the regression problem on the Weather dataset. In the classification problems, we used accuracy as metrics, while in the regression problem, we use MSE and MAE as metrics. Additionally, we apply a non-uniform quantization (3.3) in the Quantile Inform algorithm. Next, we introduce the following notation for the algorithms: Bayesian Bits as BB, Quantile Inform with uniform quantization as QIU, non-uniform Quantile Inform as QINU, and Random Bits as RB. The parameters of the algorithms are described in Appendix B.

## 5.1 CLASSIFICATION TASKS

We conducted a comparative analysis of our algorithms with Bayesian Bits (Van Baalen et al., 2020) serving as the baseline for Bayesian quantization methods. In all our experiments, we quantized the models to the same number of mean bits. Additionally, we used a non-uniform quantization (3.3) in the Quantile Inform algorithm.

For the LeNet, VGG-7, VGG-16 and ResNet-20 models we set the desired mean quantization bits equal to 4. The LeNet model was trained on the MNIST dataset. The VGG-7, VGG-16 and ResNet-20 were trained on the CIFAR-10 dataset. The training parameters for these models and their Bayesian variants are provided in Appendix B.1. The experimental results for these models are

presented in Tables 1, 2 and 3. Additional experiments on the quantization of the ResNet-20 model on the CIFAR-100 dataset are presented in Table 5 in AppendixD.

Table 1: Results of our quantization methods on LeNet and VGG-7 compared to TWN (Li et al., 2016), LR-Net (Shayer et al., 2017), RQ (Louizos et al., 2018), WAGE (Wu et al., 2018) and BB (Van Baalen et al., 2020)

| ALGORITHM | LENET (MNIST) | | VGG-7 (CIFAR-10) | |
|---|---|---|---|---|
| | MEAN BIT | ACCURACY | MEAN BIT | ACCURACY |
| TWN | 2 | 99.35 | 2 | 92.56 |
| LR-NET | 1.7 | 99.47 | 2.4 | 93.18 |
| RQ | 2 | 99.37 | 8 | 93.80 |
| WAGE | 2 | 99.60 | 2 | 93.22 |
| BB | 4 | 96.40 | 4.01 | 83.42 |
| QIU (OUR) | 3.17 | 99.05 | 4.45 | 91.51 |
| QINU (OUR) | 3.17 | 98.44 | 5 | 80.37 |
| RB (OUR) | 3.95 | 98.75 | 4.12 | 91.21 |
| FLOAT MODEL | 32 | 99.23 | 32 | 93.7 |

Table 2: Results of our quantization methods on VGG-16 and CIFAR-10 compared to BB

| ALGORITHM | MEAN BIT | ACCURACY |
|---|---|---|
| BB | 3.28 | 70.10 |
| QIU (OUR) | 4.02 | 91.45 |
| QINU (OUR) | 4.02 | 88.00 |
| RB (OUR) | 4.01 | 91.61 |
| FLOAT MODEL | 32 | 91.94 |

Table 3: Results of our quantization methods on ResNet-20 compared to LQ-Nets (Zhang et al., 2018), HAWQ (Yao et al., 2021), PACT (Choi et al., 2018), Dorefa (Zhou et al., 2016) and AdaBin (Tu et al., 2022)

| ALGORITHM | MEAN BIT | ACCURACY |
|---|---|---|
| LQ-NETS | 3 | 92.00 |
| HAWQ | 2.7 | 92.22 |
| PACT | 3 | 91.10 |
| DOREFA | 3 | 89.90 |
| ADABIN | 1 | 88.20 |
| QIU (OUR) | 3.91 | 87.65 |
| QINU (OUR) | 3.91 | 79.69 |
| RB (OUR) | 4.01 | 90.72 |
| FLOAT MODEL | 32 | 92.98 |

The performance of the proposed algorithms is comparable to that of other quantization methods. Notably, our methods QIU, QINU, and RB, which are also based on Bayesian neural networks like BB, demonstrate superior performance in several cases. For instance, on LeNet, QIU achieves 99.05% accuracy with 3.17 bits, outperforming BB with 96.40% accuracy for 4 bits. Similarly, on VGG-7, our RB method achieves 91.21% accuracy with 4.12 bits, closely matching the floating-point model with 93.7%.

For the VGG-16 model, our methods demonstrated a significant improvement over BB. QIU achieved 91.45% accuracy with an average bit-width of 4.02, compared to BB with 70.10% accuracy and 3.28 bits. For ResNet-20, our algorithms achieve results that are comparable to those of other leading quantization methods.

## 5.2 REGRESSION TASK

We performed a comparative analysis on the regression task. We used the BB algorithm on the Weather dataset from Yandex Shifts Challenge 2021 (Malinin et al., 2021) with a four-layer perceptron as the baseline. The MLP model is structured as 123FC - 512FC - 256FC - 64FC, where each "FC" denotes a fully connected layer with ReLU activation, containing 123, 512, 256, and 64 hidden units, respectively. The training parameters for the MLP model and its Bayesian variant are provided in Appendix B.2 The results of quantization of this model to 8 mean bit-width by algorithms are shown in Table 4.

Table 4: Results of our quantization methods on MLP and Weather compared to BB

| ALGORITHM | MEAN BIT | MAE | MSE |
|:---:|:---:|:---:|:---:|
| BB | 8.1 | $1.8 \cdot 10^{-2}$ | $5.98 \cdot 10^{-4}$ |
| QIU (OUR) | 6.00 | $3.1 \cdot 10^{-2}$ | $1.39 \cdot 10^{-3}$ |
| QINU (OUR) | 6.00 | $1.8 \cdot 10^{-2}$ | $6.07 \cdot 10^{-4}$ |
| RB (OUR) | 6.92 | $2.10 \cdot 10^{-2}$ | $7.46 \cdot 10^{-4}$ |
| FLOAT MODEL | 32 | $1.7 \cdot 10^{-2}$ | $5.95 \cdot 10^{-4}$ |

In the regression task on the Weather dataset using an MLP model, our methods demonstrate results comparable to those of BB. Although, BB achieves slightly better mean absolute error (MAE) and mean squared error (MSE) values, our QINU method performs almost as well, with an MAE of $1.8 \cdot 10^{-2}$ and MSE of $6.07 \cdot 10^{-4}$ using only 6 bits on average, compared to BB with 8.1 bits. This demonstrates that our methods provide a more efficient quantization while maintaining high accuracy, even in regression tasks.

## 6 DISCUSSION AND CONCLUSIONS

To address the challenge of adaptive quantization, we introduced two novel algorithms based on uncertainty modeling in Bayesian neural networks. Our results demonstrate that these algorithms effectively reduce model size while maintaining low quantization error, all within a single pass and with minimal computation after the initial training phase.

A key advantage of our method is that it follows a post-training quantization paradigm while delivering results comparable to quantization-aware training approaches. This makes it particularly suitable for quantizing pre-trained models without requiring additional extensive training. Our method also offers flexible and efficient quantization by allowing neural network layers to be quantized to any discrete bit-width, resulting in reduced computational requirements.

In conclusion, our proposed approach advances model compression techniques by providing a computationally efficient and adaptable quantization method. The demonstrated performance and flexibility suggest promising avenues for further research and practical applications in resource-constrained machine learning environments.

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

## A RANDOM BITS

### A.1 DERIVATION OF THE FIRST LOSS FUNCTION

Instead of maximizing the full likelihood (5), we can minimize its negative logarithm:

$$L(\boldsymbol{\Phi}) = -\ln p(\mathcal{M}_{\text{quant}}) = -\ln \mathbb{E}_{p(\mathbf{b})}[p(\mathcal{M}_{\text{quant}}|\mathbf{b})] \to \min_{\boldsymbol{\Phi}}.$$

However, the expression $L(\boldsymbol{\Phi})$ is still difficult to optimize. Therefore, it is proposed to use Jensen's inequality and minimize the upper bound:

$$L(\boldsymbol{\Phi}) = -\ln \mathbb{E}_{p(\mathbf{b})}[p(\mathcal{M}_{\text{quant}})|\mathbf{b})] \leq -\mathbb{E}_{p(\mathbf{b})}[\ln p(\mathcal{M}_{\text{quant}})|\mathbf{b}] = \mathcal{F}(\boldsymbol{\Phi}) \to \min_{\boldsymbol{\Phi}}.$$

Let us write $\mathcal{F}(\boldsymbol{\Phi})$ in more detail:

$$\mathcal{F}(\boldsymbol{\Phi}) = -\mathbb{E}_{p(\mathbf{b})}[\ln p(\mathcal{M}_{\text{quant}}|\mathbf{b})] = -\mathbb{E}_{p(\mathbf{b})}\left[\ln \prod_{j=1}^{N} \sum_{w \in l_j} c_j(w) q(Q(\mu_w, b_j))\right] =$$

$$= -\sum_{j=1}^{N} \mathbb{E}_{p(\mathbf{b})}\left[\ln \sum_{w \in l_j} c_j(w) q(Q(\mu_w, b_j))\right] = -\sum_{j=1}^{N} \mathbb{E}_{p(b_j)}\left[\ln \sum_{w \in l_j} c_j(w) q(Q(\mu_w, b_j))\right] \leq$$

$$\leq \{\text{Jensen's inequality}\} \leq -\sum_{j=1}^{N} \mathbb{E}_{p(b_j)}\left[\sum_{w \in l_j} c_j(w) \ln q(Q(\mu_w, b_j))\right] =$$

$$= -\sum_{j=1}^{N} \mathbb{E}_{p(b_j)}\left[\sum_{w \in l_j} c_j(w)\left(\ln \frac{1}{\sqrt{2\pi\sigma_w^2}} - \frac{(Q(\mu_w, b_j) - \mu_w)^2}{2\sigma_w^2}\right)\right].$$

Note that the constants $\{\ln 1/\sqrt{2\pi\sigma_w^2}\}$ do not depend on the parameters $\boldsymbol{\Phi}$. Consequently, they can be removed. After removing them, we arrive the following expression:

$$\mathcal{L}(\boldsymbol{\Phi}) = -\sum_{j=1}^{N} \mathbb{E}_{p(b_j)}\left[\sum_{w \in l_j} c_j(w)\left(-\frac{(Q(\mu_w, b_j) - \mu_w)^2}{\sigma_w^2}\right)\right] =$$

$$= \sum_{j=1}^{N} \mathbb{E}_{p(b_j)}\left[\sum_{w \in l_j} c_j(w) \frac{(Q(\mu_w, b_j) - \mu_w)^2}{\sigma_w^2}\right] =$$

$$= \sum_{j=1}^{N} \sum_{i=1}^{32} \underbrace{\left(\sum_{w \in l_j} c_j(w) \frac{(Q(\mu_w, b_j = i) - \mu_w)^2}{\sigma_w^2}\right)}_{\tau_{j,i}} \mathbb{P}[b_j = i] = \sum_{j=1}^{N} \sum_{i=1}^{32} \tau_{j,i} \mathbb{P}[b_j = i].$$

In this manner, we obtain the expression (6). Therefore, instead of the maximization problem (5), we can solve the minimization problem (6).

### A.2 DERIVATION OF THE SECOND LOSS FUNCTION

Note that, as before, instead of maximizing the conditional likelihood (11), we can minimize its negative logarithm, for which we also find the upper bound using Jensen's inequality

$$-\ln p(\mathcal{M}_{\text{quant}}|\widehat{\mathbf{b}}) = -\sum_{j=1}^{N} \ln\left(\sum_{w \in l_j} c_j(w) q\left(Q(\mu_w, \widehat{b}_j)\right)\right) \leq$$

$$\leq \{\text{Jensen's inequality}\} \leq -\sum_{j=1}^{N} \sum_{w \in l_j} c_j(w) \ln q\left(Q(\mu_w, \widehat{b}_j)\right) =$$

$$= -\sum_{j=1}^{N} \sum_{w \in l_j} c_j(w)\left(\ln \frac{1}{\sqrt{2\pi\sigma_w^2}} - \frac{(Q(\mu_w, \widehat{b}_j) - \mu_w)^2}{2\sigma_w^2}\right) \to \min_{\boldsymbol{\alpha}, \boldsymbol{\beta}}.$$

The constants $\{\ln 1/\sqrt{2\pi\sigma_w^2}\}$ do not depend on the parameters $\boldsymbol{\alpha} = (\alpha_1, ..., \alpha_N)$ and $\boldsymbol{\beta} = (\beta_1, ..., \beta_N)$. By removing these constants, we formulate the following expression for the second loss function

$$\mathcal{T}(\boldsymbol{\alpha}, \boldsymbol{\beta}) = \sum_{j=1}^{N} \sum_{w \in l_j} c_j(w) \frac{(Q(\mu_w, \widehat{b}_j) - \mu_w)^2}{\sigma_w^2}.$$

### A.3 DERIVATIVES OF THE SECOND LOSS FUNCTION

Next, we derive $\partial\mathcal{T}(\boldsymbol{\alpha}, \boldsymbol{\beta})/\partial\boldsymbol{\alpha}$ and $\partial\mathcal{T}(\boldsymbol{\alpha}, \boldsymbol{\beta})/\partial\boldsymbol{\beta}$ derivatives. First, we calculate the derivatives of the with respect to $\beta_j$:

$$\frac{\partial\mathcal{T}}{\partial\beta_j} = 2 \sum_{w \in l_j} c_j(w) \frac{Q(\mu_w, \widehat{b}_j) - \mu_w}{\sigma_w^2} \frac{\partial Q(\mu_w, \widehat{b}_j)}{\partial\beta_j}.$$

We need to calculate $\partial Q(\mu_w, \widehat{b}_j)/\partial\beta_j$. To do this we use the straight-through estimator approach, i.e. ignore the rounding operation:

$$\frac{\partial}{\partial\beta_j} \text{clip}\left(\frac{\mu_w - \alpha_j}{s(\widehat{b}_j)}\right) \approx \frac{\partial}{\partial\beta_j}\left(\frac{\mu_w - \alpha_j}{s(\widehat{b}_j)}\right).$$

Hence, we arrive at the following equation:

$$\frac{\partial Q(\mu_w, \widehat{b}_j)}{\partial\beta_j} = \frac{\partial}{\partial\beta_j}\left[\alpha_j + s(\widehat{b}_j)\text{clip}\left(\frac{\mu_w - \alpha_j}{s(\widehat{b}_j)}\right)\right] = \frac{\partial s(\widehat{b}_j)}{\partial\beta_j}\text{clip}\left(\frac{\mu_w - \alpha_j}{s(\widehat{b}_j)}\right) +$$

$$+ s(\widehat{b}_j)\frac{\partial}{\partial\beta_j}\text{clip}\left(\frac{\mu_w - \alpha_j}{s(\widehat{b}_j)}\right) \approx \frac{\partial s(\widehat{b}_j)}{\partial\beta_j}\text{clip}\left(\frac{\mu_w - \alpha_j}{s(\widehat{b}_j)}\right) + s(\widehat{b}_j)\frac{\partial}{\partial\beta_j}\left(\frac{\mu_w - \alpha_j}{s(\widehat{b}_j)}\right).$$

Let us describe the derivative $\partial Q(\mu_w, \widehat{b}_j)/\partial\beta_j$ in detail:

$$\frac{\partial Q(\mu_w, \widehat{b}_j)}{\partial\beta_j} \approx \frac{\partial s(\widehat{b}_j)}{\partial\beta_j}\text{clip}\left(\frac{\mu_w - \alpha_j}{s(\widehat{b}_j)}\right) + s(\widehat{b}_j)\frac{\partial}{\partial\beta_j}\left(\frac{\mu_w - \alpha_j}{s(\widehat{b}_j)}\right) =$$

$$= \frac{\partial s(\widehat{b}_j)}{\partial\beta_j}\left(\text{clip}\left(\frac{\mu_w - \alpha_j}{s(\widehat{b}_j)}\right) - \frac{\mu_w - \alpha_j}{s(\widehat{b}_j)}\right) = \frac{1}{s(\widehat{b}_j)}\frac{\partial s(\widehat{b}_j)}{\partial\beta_j}\left(Q(\mu_w, \widehat{b}_j) - \mu_w\right).$$

Notably,

$$\frac{\partial s(\widehat{b}_j)}{\partial\beta_j} = \frac{\partial}{\partial\beta_j}\left(\frac{\beta_j - \alpha_j}{2^{\widehat{b}_j} - 1}\right) = \frac{1}{2^{\widehat{b}_j} - 1}.$$

As a result,

$$\frac{\partial\mathcal{T}}{\partial\beta_j} = \frac{2}{\beta_j - \alpha_j} \sum_{w \in l_j} c_j(w) \frac{(Q(\mu_w, \widehat{b}_j) - \mu_w)^2}{\sigma_w^2}.$$

In a similar way, one can obtain the following equation:

$$\frac{\partial\mathcal{T}}{\partial\alpha_j} = -\frac{2}{\beta_j - \alpha_j} \sum_{w \in l_j} c_j(w) \frac{(Q(\mu_w, \widehat{b}_j) - \mu_w)^2}{\sigma_w^2}.$$

## B DETAILS OF MODEL TRAINING

In all experiments, the parameters for the Random Bits algorithm were:

- Number of epochs: 1000
- Learning rate (LR): $10^{-2}$
- Regularization parameter: $3 \cdot 10^3$
- Delta: 0.2

### B.1 CLASSIFICATION TASKS

The LeNet model was trained on the MNIST dataset. The VGG-7, VGG-16 and ResNet-20 were trained on the CIFAR-10 dataset. These models were trained with the following parameters:

- Batch size: 128
- Number of epochs: 300
- Optimizer: Stochastic Gradient Descent (SGD)
- Learning rate (LR): $10^{-1}$
- Weight decay: $5 \cdot 10^{-4}$
- Momentum: 0.9
- Loss function: Cross Entropy Loss

For training the Bayesian variants of the models, the following parameters were used:

- Batch size: 128
- Number of epochs: 500
- Optimizer: Stochastic Gradient Descent (SGD)
- Learning rate (LR): $10^{-3}$
- Weight decay: $5 \cdot 10^{-4}$
- Momentum: 0.9
- Loss function: ELBO Loss
    - Criterion: Cross Entropy Loss
    - Beta parameter: $10^{-6}$

Evidence Lower BOund (ELBO) is the standard loss function for Bayesian neural networks training. This loss has the following form:

$$\text{ELBO}(q(\mathbf{w}|\boldsymbol{\lambda})) = \mathbb{E}_{q(\mathbf{w}|\boldsymbol{\lambda})}[\ln p(\mathcal{D}|\mathbf{w})] - D_{\text{KL}}(q(\mathbf{w}|\boldsymbol{\lambda})||p(\mathbf{w})) \to \max_{\boldsymbol{\lambda} \in \Lambda},$$

where $q(\mathbf{w}|\boldsymbol{\lambda})$ is posterior distribution on model weights $\mathbf{w}$, $\boldsymbol{\lambda}$ is set of parameters of this distribution, $p(\mathbf{w})$ is the prior distribution and $D_{\text{KL}}$ is Kullback–Leibler divergence.

### B.2 REGRESSION TASK

We trained an MLP model using the same parameters as in the previous section (see Appendix B.1). However, the batch size was set to 512, the loss function was MSE, and the learning rate was set to $10^{-3}$.

For training the Bayesian variant of the MLP model, the batch size was also set to 512, and the learning rate was set to $10^{-4}$.

## C NON-UNIFORM QUANTIZATION

In this section we provide another justification for the information criterion. Consider our objective of representing weights of some layer $l_j$ within $b_j$ bits. This can be achieved by selecting a set of $2^{b_j}$ values, denoted as $M_j$, and subsequently approximating each $\mu_w$ with $c_j(\mu_w) := \operatorname*{argmin}_{x \in M_j}|\mu_w - x|$.

Let us approach the task of selecting the optimal set $M_j$ from a probabilistic standpoint. Assume that $\mu_w$ and $\sigma_w$ are independently drawn from absolutely continuous distributions, given by a cumulative distribution functions $F_j$ and $G_j$, respectively. To evaluate the efficacy of our quantization, we can use the mean squared quantization error:

$$\sum_{j=1}^{N} \mathbb{E}(\mu_w^{(j)} - c_j(\mu_w^{(j)}))^2,$$

where $\mu_w^{(j)}$ is drawn from $F_j$.

As discussed in the preceding section, not all weights carry equal significance. When $\sigma_w$ is substantial, indicating uncertainty in the neural network regarding a specific weight, the preservation of the precise value of $\mu_w$ may not be a critical consideration. To address this, we incorporate $\sigma_w$ into our optimization, thereby obtaining the following refined formula:

$$\sum_{j=1}^{N} \mathbb{E}\left(\frac{\mu_w^{(j)} - c_j(\mu_w^{(j)})}{\sigma_w^{(j)}}\right)^2 \approx \sum_{j=1}^{N} \frac{1}{|l_j|} \sum_{w \in l_j} \left(\frac{\mu_w - c_j(\mu_w)}{\sigma_w}\right)^2,$$

where $\sigma_w^{(j)}$ is drawn from $G_j$.

The challenge of computing the set $M_j$ to optimize $\mathbb{E}(\mu_w^{(j)} - q_j(\mu_w^{(j)}))^2$ has been addressed in Lloyd (1982). However, this solution is impractical for real-time computation and is also burdensome to store, particularly for large values of $|M_j| = 2^{b_j}$. Consequently, we require a more straightforward method to construct $M_j$, even if it turns out to be less optimal.

Could be another way to define the quantization error:

$$\sum_{j=1}^{N} \mathbb{E}|\mu_w^{(j)} - c_j(\mu_w^{(j)})|.$$

To minimize it, consider ordered elements of $M_j$: $q_1 < q_2 < \ldots < q_{m_j}$, where $m_j = |M_j| = 2^{b_j}$. Also, consider sequence $a_0 = -\infty < a_1 < \ldots < a_{m_j} = +\infty$ such that if $t$ belongs to interval $(a_{i-1}; a_i]$, then $c(t) = q_i$. Then,

$$\mathbb{E}|\mu_w^{(j)} - c_j(\mu_w^{(j)})| = \int_{-\infty}^{+\infty} |t - c_j(t)| f_j(t) \mathrm{d}t = \sum_{i=1}^{m_j} \int_{x_{i-1}}^{x_i} |t - q_i| f_j(t) \mathrm{d}t,$$

where $f_j(t)$ is the density of $F_j$: $F_j(t) = \int_{-\infty}^{t} f_j(s) \mathrm{d}s$. As mentioned before, for finite intervals $(a_{i-1}, a_i]$, the absolute quantization error is minimized if $a_i$ is the midpoint of the interval $(q_i, q_{i+1}]$. Therefore, we can instead consider

$$\mathbb{E}|\mu_w^{(j)} - c_j(\mu_w^{(j)})| = \sum_{i=1}^{m_j} \int_{(q_i+q_{i-1})/2}^{(q_i+q_{i+1})/2} |t - q_i| f_j(t) \mathrm{d}t. \tag{14}$$

To find the values $\{q_i\}_{i=1}^{m_j}$, we need to find the extrema of (14) with respect to each $q_i$

$$\frac{\partial}{\partial q_i} \sum_{i=1}^{m_j} \int_{(q_i+q_{i-1})/2}^{(q_i+q_{i+1})/2} |t - q_i| f_j(t) \mathrm{d}t = \int_{(q_i+q_{i-1})/2}^{(q_i+q_{i+1})/2} \operatorname{sign}|t - q_i| f_j(t) \mathrm{d}t = 0.$$

The last integral is equal to zero exactly when $q_i$ is the median of the segment $((q_i + q_{i-1})/2; (q_i + q_{i+1})/2]$. This gives us the following set of equations:

$$2F_j(q_i) = F_j\left(\frac{q_{i-1} + q_i}{2}\right) + F_j\left(\frac{q_i + q_{i+1}}{2}\right),$$

$$2F_j(q_1) = F_j\left(\frac{q_1 + q_2}{2}\right),$$

$$2F_j(q_{m_j}) = F_j\left(\frac{q_{m_j-1} + q_{m_j}}{2}\right).$$

This system is non–trivial to solve; however, once solved, only one value $q_1$ needs to be recorded since a full solution can be easily restored from it.

Following Lloyd (1982), another way to find an optimal solution is through an iterative process, where one has to start with some initial choice of parameters $\{q_i\}_{i=1}^{m_j}$, $\{a_i\}_{i=0}^{m_j}$ and then repeat following two steps:

1. set $\{a_i\}_{i=0}^{m_j}$ to be of midpoints between corresponding $\{q_i\}_{i=1}^{m_j}$,
2. set $\{q_i\}_{i=1}^{m_j}$ to be of medians between corresponding $\{a_i\}_{i=0}^{m_j}$.

Each step of this algorithm can only improve the solution. A good initial estimation can be obtained through quantiles of the distribution $F_j$: $\{q_i := F^{-1}(i/m_j)\}_{i=1}^{m_j}$.

# D  ADDITIONAL EXPERIMENTS

Table 5 presents a comprehensive comparison of our quantization methods, QIU and RB, with several state-of-the-art algorithms on the ResNet-20 model trained on CIFAR-100. The results are reported for two quantization configurations: W3A3 (3-bit weights and activations) and W4A4 (4-bit weights and activations). The floating-point baseline accuracy is 70.81%.

Table 5: Results of our quantization methods for ResNet-20 on CIFAR-100 (accuracy of float model is 70.81) compared to ZeroQ (Cai et al., 2020), GDFQ (Xu et al., 2020), DSG (Zhang et al., 2021), Qimera (Choi et al., 2021), ARC (Choi et al., 2022), ARC+AIT (Zhu et al., 2021), IntraQ (Zhong et al., 2021), AdaSG (Qian et al., 2023b), AdaDFQ (Qian et al., 2023a), HAST (Li et al., 2023) and DFQ (Fan et al., 2024)

| ALGORITHM | ACCURACY | |
| --- | --- | --- |
| | W3A3 | W4A4 |
| ZEROQ | 15.38 | 58.42 |
| GDFQ | 43.87 | 63.58 |
| DSG | 25.48 | 62.36 |
| QIMERA | 46.13 | 65.10 |
| ARC | 40.15 | 62.76 |
| ARC+AIT | 41.34 | 61.05 |
| INTRAQ | 48.25 | 64.98 |
| ADASG | 52.76 | 66.42 |
| ADADFQ | 52.74 | 66.81 |
| HAST | 55.67 | 66.68 |
| DFQ | 57.03 | 66.94 |
| QIU (OUR) | 14.23 (3.13 BITS) | 54.45 (4.13 BITS) |
| RB (OUR) | 48.64 (3.27 BITS) | 62.97 (4.19 BITS) |
| FLOAT MODEL | 70.81 | |

To enhance performance and accelerate the training of Bayesian models, we initialized $\mu$ with a checkpoint from the original model. This approach reduced the required iterations from 600 to 300 while improving accuracy from 54.5% to 65.5%. Additionally, we improved quantization results in each experiment by approximately 10%.

While the QIU method struggles, particularly under the W3A3 setting, where it achieves only 14.23% accuracy due to extreme compression (mean bit-width of 3.13 bits), the RB method demonstrates competitive performance. In the W4A4 configuration, QIU improves to 54.45%, but its accuracy remains lower than that of state-of-the-art methods, highlighting the need for further refinement to enhance stability and performance in aggressive quantization scenarios.

Note that for these experiments in QIU algorithm, we used the method of aggregating the informativeness of the weights, which differs from 2. For each weight $w$ in the layer $l_j$, we can evaluate the model confidence that this weight is important for the j-th layer as follows:

$$p_w = \frac{|\mu_w|/\sigma_w}{\sum_{\widetilde{w} \in l_j} |\mu_{\widetilde{w}}|/\sigma_{\widetilde{w}}}.$$

Next, for this layer, we can find its entropy

$$\mathcal{H}_j = -\sum_{w \in l_j} p_w \log p_w.$$

The high value of entropy $\mathcal{H}_j$ means that the model is uncertain and does not know which weights are most important for this layer. Note that the maximum value of entropy: $\max \mathcal{H}_j = \log |l_j|$. Next, we move from the entropy of a layer to the confidence indicator $I_j$ of the model in this layer as

$$I_j = 1 - \frac{\mathcal{H}_j}{\max \mathcal{H}_j}. \tag{15}$$

The higher the $I_j$ value, the lower the uncertainty of $l_j$. Therefore, if a layer has a high $I_j$ value (close to 1), then a large number of quantization bit can correspond to this layer. In Figure 3 we present a comparison of the informativeness of ResNet-20 layers for different datasets and aggregation methods. For CIFAR-100 aggregation using averaging gives the internal layers of the model approximately the same informativeness, which complicates the work of the algorithm. Aggregation using entropy avoids this problem.

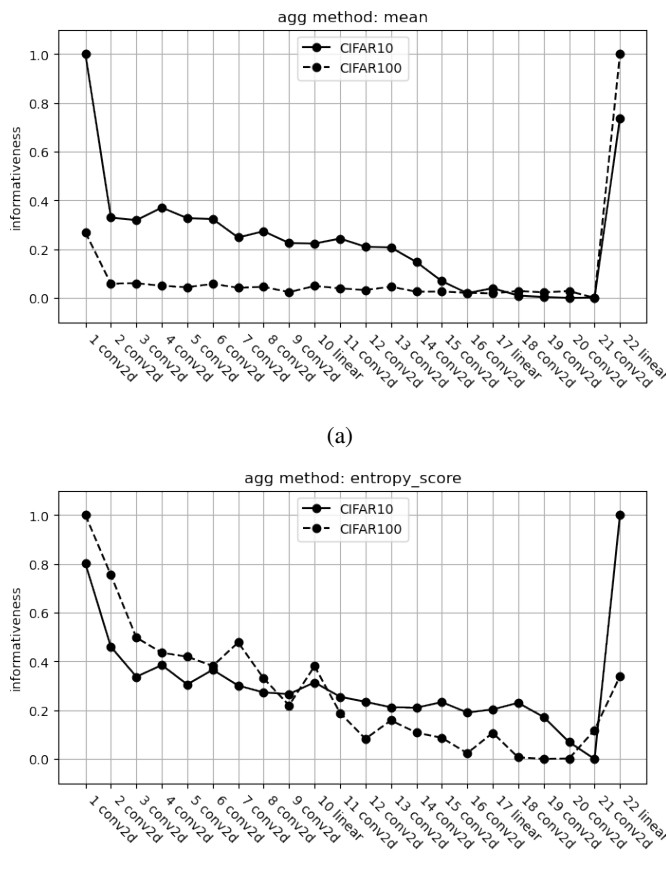

(a)

(b)

Figure 3: Comparison of layer informativeness (ResNet-20) for different aggregation approaches: (a) corresponds to the averaging method 2, (b) corresponds to the entropy method 15.

In contrast, the RB method shows strong results, achieving 48.64% accuracy under the W3A3 setting. This performance is comparable to methods like IntraQ (48.25%) and surpasses others such as GDFQ (43.87%) and DSG (25.48%). For the W4A4 configuration, RB achieves 62.97%, remaining competitive with leading methods like DFQ (66.94%) and AdaDFQ (66.81%). These results demonstrate that RB effectively balances compression and accuracy, making it a robust choice for memory-constrained environments.

# E  COMPUTATIONAL AND MEMORY COSTS OF APPROACH

## E.1  ESTIMATION OF ALGORITHMS COMPLEXITY

Next, we will evaluate the complexity of the approach relative to the number of $M$ restarts of the experiment. Then our approach with Random Bits or Quantile Inform algorithms have a computational complexity of $O(1)$, since it is enough to train the Bayesian model only once to use them. The HAWQ quantization approach also has a complexity of $O(1)$, since HAWQ analyzes the spec-

tral properties of the matrix Hesse, which does not require training restarts to calculate. Adaptive quantization based on reinforcement learning, it has a complexity of $O(M)$ due to the stochastic properties of this quantization.

## E.2 ESTIMATION OF MEMORY COSTS

Next, we will estimate the memory costs of quantization approach relative to the number of parameters $N$ of the model. Out approach with Random Bits and Quantile Inform algorithms require a trained Bayesian model that contains $2N$ parameters (mathematical expectations and variances of the weights), therefore the memory costs will be $O(N)$. The HAWQ approach does not explicitly store the Hesse matrix, but analyzes some features of the Hessian spectrum, for example, the top eigenvalue or the Hessian trace, which can be approximated without quadratic memory costs. Therefore, the total memory cost of the HAWQ algorithm is $O(N)$. For quantization based on reinforcement learning it will require $O(N)$ memory costs.

## E.3 SCALABILITY OF APPROACH

Our approach scales linearly in terms of memory consumption depending on the number of parameters. The time complexity of our method increases linearly with increasing dataset.

# F COMPUTATIONAL AND MEMORY COSTS OF QUANTIZATION ALGORITHMS

## F.1 TIME COMPLEXITY OF RANDOM BITS

Next, we will obtain an estimate of the computational complexity of one iteration of Random Bits algorithm. To do this, we first find the time complexity of calculating the coefficients $\tau_{j,i}$:

$$\tau_{j,i} = \sum_{w \in l_j} c_j(w) \frac{(Q(\mu_w, b_j = i) - \mu_w)^2}{\sigma_w^2}.$$

Since $c_j(w)$ is a constant, and the quantization calculation is performed with a single complexity, then $T(\tau_{j,i}) = O(|l_j|)$. Let's denote $\Delta b = b_{\max} - b_{\min}$. Then the calculation of all elements of the $j-$th row of the matrix $\{\tau_{j,i}\}$ has complexity: $O(|l_j|\Delta b)$.

Further, the calculation of all elements of the matrix has complexity: $T_1 = \sum_j O(|l_j|\Delta b) = O(\sum_j |l_j|\Delta b) = O(|W|\Delta b)$, where $|W|-$ the number of all weights of the model.

Next, let's assume that the calculation of probabilities $\mathbb{P}[b_j = i] = \frac{e^{\phi_{j,i}}}{\sum_s e^{\phi_{j,s}}}$ is performed for a constant. To do this, it is enough to pre-calculate the amount $\sum_s e^{\phi_{j,s}}$ for each $j$.

Now let's estimate the time complexity of calculating the first loss function:

$$\mathcal{L}(\mathbf{\Phi}) = \overbrace{\sum_{j=1}^{N} \sum_{i=1}^{32} \tau_{j,i}\mathbb{P}[b_j = i]}^{\text{quantization error}} + \underbrace{\eta(\mathbb{E}_{p(\mathbf{b})}[b^*] - \gamma)^2}_{\text{regularizer}} \to \min_{\mathbf{\Phi}}, \qquad (16)$$

where $N$ is the number of layers of the neural network.

The complexity of calculating the quantization error is $O(N\Delta b)$, since the coefficients $\{\tau_{j,i}\}$ have already been calculated, and the calculation of probabilities is performed for a constant.

To estimate the complexity of regularizer calculation, we need to describe its terms:

$$\mathbb{E}_{p(\mathbf{b})}[b_{\text{mean}}] = \mathbb{E}_{p(\mathbf{b})}\left[\frac{1}{|W|} \sum_{j=1}^{N} b_j|l_j|\right] = \frac{1}{|W|} \sum_{j=1}^{N} |l_j|\mathbb{E}_{p(b_j)}[b_j] =$$

$$= \frac{1}{|W|} \sum_{j=1}^{N} |l_j| \sum_{k=1}^{32} k \cdot \mathbb{P}[b_j = k]. \qquad (17)$$

Therefore, the calculation of $\mathbb{E}_{p(\mathbf{b})}[b_{\mathrm{mean}}]$ has complexity is $O(N\Delta b)$. Squaring is performed for a constant, so the final complexity of the regularizer is $O(N\Delta b)$. Therefore, the full complexity of calculating the function $\mathcal{L}(\mathbf{\Phi})$ is $T_2 = O(N\Delta b)$.

Calculations of the bits $\widehat{\mathbf{b}} = (\widehat{b}_1, ..., \widehat{b}_N)$ for each layer $l_j$ according to the formula:

$$\widehat{b}_j = \mathrm{round}(\mathbb{E}_{p(b_j)}[b_j]) = \mathrm{round}\left(\sum_{k=1}^{32} k \cdot \mathbb{P}[b_j = k]\right),$$

have complexity is $O(N\Delta b)$.

Next, calculation of average quantization bit $b_{\mathrm{mean}} \leftarrow \frac{1}{|W|} \sum_{j=1}^{N} |l_j| \bar{b}_j$ has compexity is $O(N)$. Therefore, the final complexity of the calculations is $T_3 = O(N\Delta b)$.

At each external iteration of Random Bits algorithm, there is an additional internal loop. Next, we will evaluate its time complexity. Random Bits algorithm is supplied with the parameter $\mathcal{K}$ - the number of internal iterations, per each of which needs to calculate a second loss function

$$\mathcal{T}(\boldsymbol{\alpha}, \boldsymbol{\beta}) = \sum_{j=1}^{N} \sum_{w \in l_j} c_j(w) \frac{(Q(\mu_w, \widehat{b}_j, \alpha_j, \beta_j) - \mu_w)^2}{\sigma_w^2}.$$

Since each term $\mathcal{T}(\boldsymbol{\alpha}, \boldsymbol{\beta})$ is calculated as a constant, the complexity of its calculation is $O(|W|)$.

Next, we estimate the complexity of calculating the approximate gradients of the function $\mathcal{T}(\boldsymbol{\alpha}, \boldsymbol{\beta})$, which are calculated using the formulas:

$$\frac{\partial \mathcal{T}}{\partial \beta_j} \approx \frac{2}{\beta_j - \alpha_j} \sum_{w \in l_j} c_j(w) \frac{(Q(\mu_w, \bar{b}_j) - \mu_w)^2}{\sigma_w^2},$$

$$\frac{\partial \mathcal{T}}{\partial \alpha_j} \approx -\frac{2}{\beta_j - \alpha_j} \sum_{w \in l_j} c_j(w) \frac{(Q(\mu_w, \bar{b}_j) - \mu_w)^2}{\sigma_w^2}.$$

Calculating these gradients for an arbitrary layer $l_j$ has a complexity of $O(|l_j|)$. Hence, the total complexity of calculating all gradients is $O(|W|)$.

Therefore, the complexity of one internal iteration is $O(|W|)$. Since the inner iterations are $\mathcal{K}$, the total complexity of the entire inner loop is $T_4 = O(\mathcal{K}|W|)$.

Next, to estimate the complexity of the calculation of $\nabla \mathcal{L}(\mathbf{\Phi})$ (16), we first estimate the complexity of calculating the probability derivative in two different cases:

$$\text{if k} \neq \text{i:} \ \frac{\partial \mathbb{P}[b_j = i]}{\partial \phi_{j,k}} = \frac{\partial}{\partial \phi_{j,k}} \frac{e^{\phi_{j,i}}}{\sum_{s=1}^{32} e^{\phi_{j,s}}} = -\frac{e^{\phi_{j,i} + \phi_{j,k}}}{\left(\sum_{s=1}^{32} e^{\phi_{j,s}}\right)^2};$$

$$\text{if k} = \text{i:} \ \frac{\partial \mathbb{P}[b_j = k]}{\partial \phi_{j,k}} = \frac{\partial}{\partial \phi_{j,k}} \frac{e^{\phi_{j,k}}}{\sum_{s=1}^{32} e^{\phi_{j,s}}} = \frac{e^{\phi_{j,k}}}{\sum_{s=1}^{32} e^{\phi_{j,s}}} - \frac{e^{2\phi_{j,k}}}{\left(\sum_{s=1}^{32} e^{\phi_{j,s}}\right)^2}.$$

Since we have previously calculated the value of $\sum_{s=1}^{32} e^{\phi_{j,s}}$, then taking this derivative has a complexity of $O(1)$.

Next, we find the complexity of the derivative of the quantization error in (16):

$$\frac{\partial}{\partial \phi_{j,k}} \sum_{t=1}^{N} \sum_{i=1}^{32} \tau_{t,i} \mathbb{P}[b_t = i] = \sum_{i=1}^{32} \tau_{j,i} \frac{\partial \mathbb{P}[b_j = i]}{\partial \phi_{j,k}} \Rightarrow \text{complexity is } O(\Delta b). \quad (18)$$

Now let's find the computational complexity for the derivative of the regularizer of the function (16). To do this, first evaluate the complexity of the derivative of the expression (17):

$$\frac{\partial \mathbb{E}_{p(\mathbf{b})}[b^*]}{\partial \phi_{j,k}} = \frac{\partial}{\partial \phi_{j,k}} \left(\frac{1}{|W|} \sum_{j=1}^{N} |l_j| \sum_{i=1}^{32} i \cdot \mathbb{P}[b_j = i]\right) = \frac{|l_j|}{|W|} \sum_{i=1}^{32} i \frac{\partial \mathbb{P}[b_j = i]}{\partial \phi_{j,k}}. \quad (19)$$

Taking into account (18), the complexity of calculating this derivative is $O(\Delta b)$. Now let's estimate the computational complexity for the regularizer:

$$\frac{\partial}{\partial \phi_{j,k}}(\mathbb{E}_{p(\mathbf{b})}[b^*] - \gamma)^2 = 2(\mathbb{E}_{p(\mathbf{b})}[b^*] - \gamma)\frac{\partial \mathbb{E}_{p(\mathbf{b})}[b^*]}{\partial \phi_{j,k}}$$

Since the value of $\mathbb{E}_{p(\mathbf{b})}[b^*] - \gamma$ was previously found, then taking into account (19) the computational complexity for the derivative of the regularizer $O(\Delta b)$.

This means that the computation of the derivative of the loss function (16) for one argument $\phi_{j,i}$ has complexity $O(\Delta b)$. Since there are only $N\Delta b$ arguments, the complexity $T_5$ of computing the full gradient $\nabla \mathcal{L}(\mathbf{\Phi})$ is equal to $O(N\Delta b^2)$.

The total complexity of Random Bits algorithm at one iteration is:

$$\begin{aligned} T_{\text{result}} = T_1 + T_2 + T_3 + T_4 + T_5 &= O(|W|\Delta b) + O(N\Delta b)+ \\ &+ O(N\Delta b) + O(\mathcal{K}|W|) + O(N\Delta b^2) = O\big(|W|(\Delta b + \mathcal{K}) + N\Delta b^2\big). \end{aligned}$$

### F.2 MEMORY COST OF RANDOM BITS

To work, Random Bits algorithm requires a trained Bayesian neural network for input, so there are the following memory costs $O(|W|)$. On the other hand, memory is required to store matrices of algorithm parameters $\mathbf{\Phi} = (\phi_1, ..., \phi_N)$ and coefficients $\{\tau_{j,i}\}$, the sizes of both of which is $N \times \Delta b$, where $N$ - number of model layers, $\Delta b$ - the number of possible bits for quantization. All additional storage pre-calculations, which reduce the time complexity of Random Bits, require less than $O(N\Delta b)$ memory costs. Hence the total memory cost of Random Bits is $O(|W| + N\Delta b)$.

### F.3 TIME COMPLEXITY OF QUANTILE INFORM

To assess the complexity of Quantile Inform algorithm, we will write its pseudocode. The algorithm allows finding suitable sets of quantization bits.

---

**Algorithm 2: Quantile Inform**

  **Input:**
    scaled informativeness of BNN layers - *infos*[ ]
    required average quantization bit rate - $\gamma$
    acceptable deviation for the average bit rate - $\delta$
  **for** $m = 1, ..., \Delta b$ **do**
    **for** $b_{\text{init}} = b_{\min}, ..., b_{\max}$ **do**
      *borders*[ ] $\leftarrow [0, 1/m, 2/m, ..., 1]$ /* m is the number of intervals    */
      calculate bits $\widehat{\boldsymbol{b}} = (\widehat{b}_1, ..., \widehat{b}_N) \leftarrow \text{getBits}(borders, infos, b_{\text{init}})$
      get mean quantization bit $b_{\text{mean}} \leftarrow \frac{1}{|W|}\sum_{j=1}^{N} |l_j|\widehat{b}_j$
      **if** $\gamma - \delta < b_{\text{mean}} \leq \gamma$ **then**
        save bits $\widehat{b}_1, ..., \widehat{b}_N$
      **end if**
    **end for**
  **end for**

---

The iteration time of the inner loop is fully estimated by the computational complexity of the function getBits. This function finds for each j-th layer the interval $\Delta_j = [borders[k], borders[k+1]]$ such that $infos[j] \in \Delta_j$. Next, the layer $l_j$ will be assigned $k + b_{\text{init}}$ bit for quantization. If the array $infos[$ ] is pre-sorted (which takes $O(N \log N)$), then the time costs of getBits is $O(\Delta b + N)$, where $N$ is the number of layers. Taking into account the nesting of the loop, the time complexity of the algorithm is $O(\Delta b^2(\Delta b + N) + N \log N)$.

### F.4 MEMORY COST OF QUANTILE INFORM

The memory cost is fully estimated by the number of model weights, i.e. $O(|W|)$.

# G  ABOUT COMPUTATIONAL OVERHEAD AND PRACTICAL IMPLEMENTATION CONSIDERATIONS

All experiments were conducted using an NVIDIA Tesla A100 GPU (40GB RAM). the training time for models are presented in Table 6.

Table 6: Training time for models (in hours)

| ARCHITECTURE | DATASET | ORIGINAL MODEL | BAYESIAN MODEL |
|---|---|---|---|
| LENET | MNIST | 0.27 | 0.46 |
| VGG-7 | CIFAR-10 | 0.92 | 2.87 |
| VGG-16 | CIFAR-10 | 1.10 | 3.36 |
| RESNET-20 | CIFAR-10 | 0.67 | 2.14 |
| RESNET-20 | CIFAR-100 | 0.98 | 4.22 |
| MLP | WEATHER | 0.04 | 0.23 |

Quantization algorithm runtime:

- QI: 20 to 120 seconds.
- RB: 1.5 to 7 minutes.

The proposed quantization methods rely on the estimation of weight importance, which is derived from the Bayesian framework. In this context, the informativeness of each weight is quantified as the ratio of its mean ($\mu$) to its variance ($\sigma^2$). This metric is essential for the algorithms, as it allows the identification of the most critical weights in the network.

To compute these values, the neural network needs to be trained as a Bayesian model, where each weight is represented by a distribution (typically Gaussian) instead of a single point estimate. This training process requires optimizing both the means ($\mu$) and variances ($\sigma^2$) of the weights, which adds complexity compared to traditional neural network training.

One way to mitigate the additional computational overhead of training a Bayesian neural network is to initialize the means ($\mu$) of the Bayesian weights using the weights of a pretrained standard model. This approach offers several advantages:

- Faster Convergence: The pretrained weights provide a well-tuned starting point for $\mu$, reducing the time required for the Bayesian network to converge during training.
- Better Performance: Starting from pretrained weights ensures that the model is initialized closer to a good solution, potentially leading to higher final accuracy.
- Reduced Variance Exploration: Since $\mu$ starts with meaningful values, the optimization focuses more on fine-tuning the variances ($\sigma^2$), further streamlining the process.

To enhance performance and accelerate the training of Bayesian models, we performed an experiment with ResNet-20 on CIFAR-100, initializing $\mu$ with a checkpoint from the original model. This approach reduced the required iterations from 600 to 300 while improving accuracy from 54.5% to 65.5%. Additionally, we improved quantization results in each experiment by approximately 10%.

This initialization technique effectively leverages the progress made during the training of the original model, making the transition to the Bayesian variant more practical for large-scale applications.

