# OpenReview forum: "Neural Network Adaptive Quantization based on Bayesian Deep Learning"
_ICLR.cc/2025/Conference — Submitted to ICLR 2025_

### Official Review · Reviewer_D6CC · 2024-10-23

**Soundness:** 2
**Presentation:** 2
**Contribution:** 2
**Rating:** 3
**Confidence:** 3

**Summary:**

The article presents a novel approach to adaptive quantization in neural networks, leveraging epistemic uncertainty analysis. By treating the quantized model as a Bayesian neural network with stochastic weights, the method utilizes mean values for weight estimation and standard deviations to indicate uncertainty—where more bits signify lower uncertainty. The authors introduce two new algorithms for mixed-precision quantization: QuantileInform, which allocates bit-width based on uncertainty across layers, and RandomBits, which uses stochastic gradient-based optimization to enhance quantization likelihood.

**Strengths:**

1.Innovative Approach: The integration of epistemic uncertainty analysis into quantization offers a fresh perspective on optimizing neural networks.

2.Mixed-Precision Algorithms: The introduction of two new algorithms allows for flexible bit-width allocation, enhancing model efficiency.

**Weaknesses:**

1.Limited Experimental Scope: Lacks experiments on Transformer and larger datasets like ImageNet.

2.Lack of Benchmarking: No comparison with state-of-the-art methods obscures performance insights,like Adabin.

3.The actual acceleration effect is not shown in the experiment.

**Questions:**

1. Does your method only quantize the weights?

2.Why introduce uncertainty to quantize models? What are the advantages?

3.How do you ensure that your entire algorithm can be implemented on low-bitwidth hardware?

---

> ### Author Response · Authors · 2024-11-25
>
> > 1. Does your method only quantize the weights?
>
> In this work, we focused solely on quantizing the weights of the model to validate the idea of using informativeness from Bayesian models. This allowed us to simplify the problem and confirm the effectiveness of our approach, which yielded promising results.
>
> In future research, we plan to extend our methods to include activation quantization. Incorporating activation quantization could potentially further improve the trade-off between model accuracy and compression.
>
> It is also worth noting that several post-training quantization (PTQ) methods achieve excellent results by quantizing only the weights, without altering activations. For example, OPTQ [1], a PTQ algorithm specifically designed for generative pre-trained transformers (GPTs), achieves accurate and efficient quantization by focusing exclusively on weight quantization. This demonstrates that weight quantization alone can already provide significant efficiency gains in large-scale models like LLMs.
>
> [1] Frantar, E., Ashkboos, S., Hoefler, T., & Alistarh, D. (2022). OPTQ: Accurate quantization for generative pre-trained transformers. The Eleventh International Conference on Learning Representations.
> https://openreview.net/forum?id=tcbBPnfwxS

---

> ### Author Response · Authors · 2024-11-26
>
> > 1. Limited Experimental Scope: Lacks experiments on Transformer and larger datasets like ImageNet.
>
> Modern models, such as LLMs, are primarily based on transformers, which consist of attention and fully connected layers. Recently, authors in [1] highlighted that the key aspect of transformer performance lies in optimizing fully connected layers. Additionally, hybrid architectures like conformers, which combine transformers with convolutional layers, have been proposed in [2]. In our work, we propose methods that effectively quantize architectures with both fully connected and convolutional layers, demonstrating their applicability to transformers and conformers.
>
> In response to the reviewer’s comment, we are preparing additional comparisons with other quantization algorithms, which we plan to include in the revised manuscript. Regarding the choice of dataset, we conducted experiments on CIFAR-100 instead of ImageNet to reduce computational complexity. CIFAR-100 provides comparable diversity and complexity to ImageNet, as demonstrated in [3], while being more manageable in terms of experimental setup.Additionally, the level of label errors in CIFAR-100 is similar to that of ImageNet, as shown in [4], ensuring that our results remain meaningful and robust.
>
> [1] He, S., Sun, G., Shen, Z., & Li, A. (2024). What Matters in Transformers? Not All Attention is Needed. https://arxiv.org/abs/2406.15786
>
> [2] Gulati, A., Qin, J., Chiu, C.-C., Parmar, N., Zhang, Y., Yu, J., Han, W., Wang, S., Zhang, Z., Wu, Y., & Pang, R. (2020). Conformer: Convolution-augmented Transformer for Speech Recognition. https://arxiv.org/abs/2005.08100
>
> [3] Darlow, L. N., Crowley, E. J., Antoniou, A., & Storkey, A. J. (2018). Cinic-10 is not imagenet or cifar-10. arXiv preprint arXiv:1810.03505. https://arxiv.org/abs/1810.03505
>
> [4] Northcutt, C. G., Athalye, A., & Mueller, J. (2021). Pervasive label errors in test sets destabilize machine learning benchmarks. arXiv preprint arXiv:2103.14749. https://arxiv.org/abs/2103.14749

---

> ### Author Response · Authors · 2024-11-26
>
> > 3. The actual acceleration effect is not shown in the experiment.
>
> > 3. How do you ensure that your entire algorithm can be implemented on low-bitwidth hardware?
>
> In our work, we focused on reducing the memory required for storing model weights through quantization. The quantization process in our approach involves training a Bayesian variant of the neural network to estimate weight informativeness, followed by selecting mixed-precision quantization configurations for the layers. This process requires iterative optimization and is computationally intensive, making it better suited for server-side preparation rather than direct implementation on low-bitwidth hardware.
>
> In our experiments, we simulated quantization effects by storing weights in float32, allowing us to evaluate the accuracy of the quantized models without directly using low-bitwidth hardware. As a result, we did not measure acceleration effects or test deployment on hardware explicitly.
>
> Given the nature of our method, its direct adaptation to low-bitwidth hardware would require significant modifications to handle the Bayesian training and mixed-precision optimization steps efficiently. Therefore, while our method ensures compatibility with standard quantization formats used in low-bitwidth hardware, its primary contribution lies in proposing an effective server-side quantization framework rather than a hardware-specific implementation.

---

> ### Author Response · Authors · 2024-11-26
>
> > 2. Lack of Benchmarking: No comparison with state-of-the-art methods obscures performance insights,like Adabin.
>
> In response to the reviewer’s comment, we are preparing additional comparisons with other quantization algorithms, which we plan to include in the revised manuscript.
>
> While AdaBin [1] is specifically designed for 1-bit quantization (binarization), which differs in focus from our approach of mixed-precision quantization, we recognize its relevance as a benchmark. To address this, we will include the results of AdaBin applied to ResNet-20 on CIFAR-10 in Table 3 of the revised manuscript. This will provide a clearer comparison of our method's performance against a well-established 1-bit quantization algorithm.
>
> [1] Tu, Z., Chen, X., Ren, P., & Wang, Y. (2022, October). Adabin: Improving binary neural networks with adaptive binary sets. In European conference on computer vision (pp. 379-395). Cham: Springer Nature Switzerland.

---

> ### Author Response · Authors · 2024-12-02
>
> > 2.Why introduce uncertainty to quantize models? What are the advantages?
>
> The introduction of uncertainty in neural network weights provides a natural way to evaluate the importance of accurately representing these weights without incurring additional memory or computational costs during training. Higher uncertainty indicates lower sensitivity to precise weight values, enabling the use of fewer bits for quantized representations without compromising performance.
>
> By leveraging uncertainty estimates from Bayesian neural networks, rather than alternative approaches like deep ensembles, our method avoids the significant memory overhead typically associated with such techniques, ensuring a more efficient training process. This makes the quantization process not only more adaptive but also computationally and memory efficient.

---

> ### Author Response · Authors · 2024-12-02
>
> We have updated our paper by adding additional experiments in the appendix, comparing the quantization results of ResNet-20 on the CIFAR-100 dataset with other state-of-the-art algorithms. These experiments provide a more comprehensive evaluation of our methods under low-bit settings.
>
> In addition, the appendix now includes detailed information on the computational complexity, memory requirements, and implementation complexity of the proposed algorithms.
>
> We believe these updates offer readers a clearer understanding of the potential, efficiency, and practical applicability of our methods in scenarios requiring efficient neural network compression.

---

### Official Review · Reviewer_JwPr · 2024-10-30

**Soundness:** 2
**Presentation:** 2
**Contribution:** 2
**Rating:** 3
**Confidence:** 5

**Summary:**

The paper introduces algorithms that leverage uncertainty analysis in Bayesian Neural Networks (BNNs) to assess the importance of weights and apply it into mixed-precision quantization to determine optimal bit-width of each layer.

**Strengths:**

The paper propose a mixed-precision quantization, achieving comparable performance compared to traditional Quantization-Aware Training (QAT) algorithms.

The concepts are well-explained.

**Weaknesses:**

1. The approach builds on well-established methods, particularly drawing from prior work [1] on measuring weight importance in pruning and applying it to quantization. This reduces the novelty of the contribution, as the techniques have been previously discussed in [1]. However, the paper does extend this work effectively into the domain of mixed-precision quantization, applying the importance metrics in a novel context. Despite this extension, the core idea remains closely tied to existing pruning methods, which limits the originality of the contribution.

[1] Blundell, Charles, et al. "Weight uncertainty in neural network." International conference on machine learning. PMLR, 2015.

2. The proposed algorithms lack sufficient evaluation on state-of-the-art models, making it challenging to assess how powerful they are in real-world applications. Without benchmarking against modern models such as LLMs or using more diverse and complex datasets, it becomes difficult to measure the true effectiveness and robustness of the approach.
In addition, a comparison with the latest quantization algorithms is necessary.

3. The method relies heavily on BNNs, making it unsuitable for models that are not trained within a BNN, thereby restricting its applicability.

**Questions:**

See weakness.

---

> ### Author Response · Authors · 2024-11-21
>
> > 2. The proposed algorithms lack sufficient evaluation on state-of-the-art models, making it challenging to assess how powerful they are in real-world applications. Without benchmarking against modern models such as LLMs or using more diverse and complex datasets, it becomes difficult to measure the true effectiveness and robustness of the approach. In addition, a comparison with the latest quantization algorithms is necessary.
>
> Modern models, such as LLMs, are primarily based on transformers,  which consist of attention and fully connected layers. Recently, authors in [1] highlighted that the key aspect of transformer performance lies in optimizing fully connected layers. Additionally, hybrid architectures like conformers, which combine transformers with convolutional layers, have been proposed in [2]. In our work, we propose methods that effectively quantize architectures with both fully connected and convolutional layers, demonstrating their applicability to transformers and conformers.
>
> In response to the reviewer’s comment, we are preparing additional comparisons with other quantization algorithms, which we plan to include in the revised manuscript.
>
> [1] He, S., Sun, G., Shen, Z., & Li, A. (2024). What Matters in Transformers? Not All Attention is Needed. https://arxiv.org/abs/2406.15786
>
> [2] Gulati, A., Qin, J., Chiu, C.-C., Parmar, N., Zhang, Y., Yu, J., Han, W., Wang, S., Zhang, Z., Wu, Y., & Pang, R. (2020). Conformer: Convolution-augmented Transformer for Speech Recognition. https://arxiv.org/abs/2005.08100

---

> ### Author Response · Authors · 2024-11-24
>
> > 1. The approach builds on well-established methods, particularly drawing from prior work [1] on measuring weight importance in pruning and applying it to quantization. This reduces the novelty of the contribution, as the techniques have been previously discussed in [1]. However, the paper does extend this work effectively into the domain of mixed-precision quantization, applying the importance metrics in a novel context. Despite this extension, the core idea remains closely tied to existing pruning methods, which limits the originality of the contribution.
> >
> > [1] Blundell, Charles, et al. "Weight uncertainty in neural network." International conference on machine learning. PMLR, 2015.
>
> While we draw inspiration from [1] in using the ratio of the mathematical expectation of a weight to its variance as a measure of weight importance (informativity), our approach significantly diverges in its application and scope. Specifically, [1] addresses the problem of neural network pruning, whereas our work tackles the distinct challenge of mixed-precision quantization.
>
> The QuantileInform algorithm extends the concept of weight informativeness by aggregating it at the layer level to assess the importance of each layer. This importance metric is then utilized to guide the quantization process, specifically by searching for optimal combinations of bit widths for each layer, thereby making the search process more efficient.
>
> The Random Bits algorithm, on the other hand, is based on an entirely different principle. It maximizes the full likelihood of the model after quantization. In this algorithm, weight informativeness is used solely to compute coefficients in a linear combination of probability densities, approximating the conditional likelihood of weights in a layer. Notably, these coefficients could be derived through other means, making weight informativeness only a minor component of the Random Bits algorithm.
>
> Thus, while our work builds on an idea from [1], it introduces novel methodologies for quantization and extends the application of importance metrics into a fundamentally new domain.
>
> [1] Blundell, Charles, et al. "Weight uncertainty in neural network." International conference on machine learning. PMLR, 2015.

---

> ### Comment · Reviewer_JwPr · 2024-11-26
>
> Thank you for your effort to address my concern. However, the current experimental results without comparisons to state-of-the-art algorithms on the latest models appears to be at the level of a proof of concept, making it challenging to fully evaluate the impact and applicability of your work. Thus, I maintain my current evaluation.

---

> ### Author Response · Authors · 2024-12-02
>
> > The method relies heavily on BNNs, making it unsuitable for models that are not trained within a BNN, thereby restricting its applicability.
>
> All weights in the Bayesian neural network are represented by probability distributions over possible values, while the expectations correspond to the most likelihood values. Accordingly, it is correct to initialize the expectations with point estimates of the weights of a pre-trained non-Bayesian neural network and to fine-tune the distribution over the weights of the neural network. We also show the effectiveness of the described transformation of a non-Bayesian neural network into a Bayesian one experimentally in the revised version of our paper.

---

> ### Author Response · Authors · 2024-12-02
>
> We have updated our paper by adding additional experiments in the appendix, comparing the quantization results of ResNet-20 on the CIFAR-100 dataset with other state-of-the-art algorithms. These experiments provide a more comprehensive evaluation of our methods under low-bit settings.
>
> In addition, the appendix now includes detailed information on the computational complexity, memory requirements, and implementation complexity of the proposed algorithms.
>
> We believe these updates offer readers a clearer understanding of the potential, efficiency, and practical applicability of our methods in scenarios requiring efficient neural network compression.

---

### Official Review · Reviewer_GaGi · 2024-11-03

**Soundness:** 4
**Presentation:** 3
**Contribution:** 3
**Rating:** 6
**Confidence:** 3

**Summary:**

The paper introduces a novel approach to solve the adaptive quantization problem in neural networks based on Bayesian neural networks and uncertainty analysis. Two new algorithms are presented: QuantileInform, which utilizes uncertainty to allocate bit-width across layers based on informativeness, and RandomBits, which employs stochastic gradient-based optimization techniques to maximize the full likelihood of quantization. The paper demonstrates the effectiveness of these approaches through experiments on MNIST and CIFAR-10 datasets, showing that the methods can achieve comparable or better accuracy than existing quantization methods while using fewer bits. For instance, they reduce the average bit-width of the VGG16 model to 3.05 with 90.5% accuracy on CIFAR-10 compared to 91.9% for the non-quantized model.

**Strengths:**

The paper demonstrates several notable strengths in its approach to neural network quantization. A key strength is the novel utilization of Bayesian neural networks to determine optimal bit-width allocation, departing from traditional fixed-width quantization methods. The authors present a mathematically rigorous foundation for their approach, with clear derivations and well-justified methodological choices. Their innovative interpretation of bits as random discrete variables enables the application of gradient optimization techniques, representing a creative solution to the optimization challenge. The presentation is particularly strong, with clear mathematical notation and well-designed figures that effectively communicate both methodology and results. The experimental results show promising performance on the MNIST and CIFAR-10 datasets, demonstrating the practical utility of their approach in reducing model size while maintaining accuracy. The paper's thorough treatment of the theoretical underpinnings, combined with practical implementation considerations, provides a comprehensive view of the proposed solution. Additionally, the authors' explicit acknowledgment of limitations and potential areas for future work demonstrates scientific rigor and transparency in their research approach.

**Weaknesses:**

Despite its strengths, the paper has several areas that could be improved. A significant limitation is the restricted scope of experimental validation, with testing limited to MNIST and CIFAR-10 datasets. More extensive evaluation on larger, more complex datasets would be necessary to fully demonstrate the scalability and robustness of the proposed algorithms. The authors make claims about computational efficiency and linear memory requirements compared to methods using Hessian computation, but these claims lack rigorous theoretical proof or experimental validation. The comparison with state-of-the-art methods could be more comprehensive, particularly with respect to recent approaches like AMC or HAWQ. The paper would benefit from a more detailed analysis of computational overhead and practical implementation considerations, especially for larger models and datasets. A theoretical analysis of runtime complexity is notably absent, which would be valuable for understanding the scalability of the approach. While the mathematical foundations are sound, some of the more complex derivations, particularly in the appendix, could benefit from additional clarification and simplification to improve accessibility. These limitations, while not invalidating the paper's contributions, represent opportunities for strengthening the work and its potential impact on the field.

**Questions:**

Could you provide more extensive comparisons with state-of-the-art methods like AMC and HAWQ?
What is the theoretical and practical runtime complexity of your algorithms?
How does the approach scale with larger datasets and more complex architectures?
Can you provide empirical evidence for the claimed linear memory requirements?

---

> ### Author Response · Authors · 2024-11-26
>
> > The comparison with state-of-the-art methods could be more comprehensive, particularly with respect to recent approaches like AMC or HAWQ. The paper would benefit from a more detailed analysis of computational overhead and practical implementation considerations, especially for larger models and datasets.
>
> > Could you provide more extensive comparisons with state-of-the-art methods like AMC and HAWQ?
>
> Thank you for your valuable feedback. We would like to clarify that AMC [1] is primarily an algorithm for weight pruning, whereas our work focuses on developing algorithms for quantization. This represents a fundamental difference in the scope and application of the methods.
>
> Regarding HAWQ [2], we have already included a direct comparison in Table 3 of the manuscript, presenting quantization results for ResNet-20 on the CIFAR-10 dataset. Our methods demonstrate competitive performance in this context.
>
> In response to the reviewer’s comment, we are conducting additional experiments to compare our methods with other quantization algorithms. We also plan to extend these evaluations to the CIFAR-100 dataset. These additional comparisons will be included in the revised manuscript.
>
> Furthermore, to address the reviewer’s concern about computational overhead and implementation considerations, we will include a detailed discussion in the appendix. This will cover the complexity of implementing our algorithms, the challenges of tuning hyperparameters for the Bayesian network, and the configuration of quantization algorithms. By providing this information, we aim to enhance the practical relevance and accessibility of our approach.
>
> [1] He, Y., Lin, J., Liu, Z., Wang, H., Li, L. J., & Han, S. (2018). Amc: Automl for model compression and acceleration on mobile devices. In Proceedings of the European conference on computer vision (ECCV) (pp. 784-800). https://arxiv.org/abs/1704.00763v1
>
> [2] Yao, Z., Dong, Z., Zheng, Z., Gholami, A., Yu, J., Tan, E., ... & Keutzer, K. (2021, July). Hawq-v3: Dyadic neural network quantization. In International Conference on Machine Learning (pp. 11875-11886). PMLR. https://proceedings.mlr.press/v139/yao21a.html

---

> ### Author Response · Authors · 2024-12-02
>
> > A theoretical analysis of runtime complexity is notably absent, which would be valuable for understanding the scalability of the approach.
>
> > What is the theoretical and practical runtime complexity of your algorithms?
>
> > How does the approach scale with larger datasets and more complex architectures?
>
> > The authors make claims about computational efficiency and linear memory requirements compared to methods using Hessian computation, but these claims lack rigorous theoretical proof or experimental validation.
>
> > Can you provide empirical evidence for the claimed linear memory requirements?
>
> We recognize the importance of providing a theoretical and practical analysis of runtime complexity, as well as empirical validation of our claims regarding memory efficiency. To address these concerns, we have updated our manuscript and included a dedicated section in the appendix.
>
> The complexity grows linearly with the number of parameters in the model ($O(N)$). This is due to the need to compute Bayesian weight distributions ($\mu$ and $\sigma^2$) for each parameter and perform quantization searches across all layers. Memory requirements are also linear ($O(N)$).
>
> Since our approach involves training a Bayesian neural network, the computational complexity depends on the number of samples in the dataset, scaling linearly ($O(M)$) with the dataset size. Each forward and backward pass during training processes all samples individually, which ensures that the training phase remains efficient even for larger datasets. The quantization algorithms, QI and RB, however, do not directly depend on the dataset. Their complexity is determined solely by the number of parameters in the neural network, making them independent of dataset size.
>
> By providing these detailed analyses, we aim to clarify the scalability and computational efficiency of our approach, addressing the reviewer’s concerns comprehensively.

---

> ### Author Response · Authors · 2024-12-02
>
> > A significant limitation is the restricted scope of experimental validation, with testing limited to MNIST and CIFAR-10 datasets. More extensive evaluation on larger, more complex datasets would be necessary to fully demonstrate the scalability and robustness of the proposed algorithms.
>
> In response, we have expanded our evaluation in the revised manuscript and included additional comparisons with other quantization algorithms.
>
> Beyond the experiments conducted on CIFAR-10, we have also performed tests using ResNet-20 on CIFAR-100, a dataset that offers greater complexity and diversity compared to CIFAR-10. These additional experiments demonstrate the scalability of our proposed algorithms to datasets with higher levels of variability and complexity.

---

> ### Author Response · Authors · 2024-12-02
>
> We have updated our paper by adding additional experiments in the appendix, comparing the quantization results of ResNet-20 on the CIFAR-100 dataset with other state-of-the-art algorithms. These experiments provide a more comprehensive evaluation of our methods under low-bit settings.
>
> In addition, the appendix now includes detailed information on the computational complexity, memory requirements, and implementation complexity of the proposed algorithms.
>
> We believe these updates offer readers a clearer understanding of the potential, efficiency, and practical applicability of our methods in scenarios requiring efficient neural network compression.

---

### Meta-Review · Area_Chair_5vHb · 2024-12-25

**Metareview:**

This work proposes two algorithms, QuantileInform and RandomBits, for mixed-precision quantization based on epistemic uncertainty analysis.

Though the reviewers appreciated the theoretical insights provided in the paper, the primary concern is the lack of convincing experiments. The existing experiments are on MNIST, CIFAR10, and CIFAR100 (new experiment during rebuttal) using VGG and ResNet-20 architectures. This significantly limits the applicability of the approach. The authors themselves argued that their approach is applicable to transformers and conformers, therefore, I'd strongly suggest them to show the effectiveness of their method in such models to increase the scope of their work.

Not mentioning other comments by the reviewers as the one mentioned above itself is a significant one, I recommend rejecting this work in its current form. However, given the appreciation for the insights provided by the authors, I'd strongly suggest them to work on the comments above to increase the applicability of their proposed approach.

**Additional Comments On Reviewer Discussion:**

One of the major concerns raised during the rebuttal was related to the lack of convincing experiments which I completely agree with. This aspect of the paper was not answered convincingly during the rebuttal as, understandably, this will require significant work and a bit of rewriting.

---

### Decision · Program_Chairs · 2025-01-22

Reject